# Safety and Feasibility Assessment of Repetitive Vascular Occlusion Stimulus (RVOS) Application to Multi-Organ Failure Critically Ill Patients: A Pilot Randomised Controlled Trial

**DOI:** 10.3390/jcm11143938

**Published:** 2022-07-06

**Authors:** Ismita Chhetri, Julie E. A. Hunt, Jeewaka R. Mendis, Lui G. Forni, Justin Kirk-Bayley, Ian White, Jonathan Cooper, Karthik Somasundaram, Nikunj Shah, Stephen D. Patterson, Zudin A. Puthucheary, Hugh E. Montgomery, Benedict C. Creagh-Brown

**Affiliations:** 1Intensive Care Unit, Royal Surrey County Hospital, NHS Foundation Trust, Guildford GU2 7XX, UK; i.chhetri@imperial.ac.uk (I.C.); luiforni@nhs.net (L.G.F.); jkb@nhs.net (J.K.-B.); 2Faculty of Health and Medical Sciences, School of Biosciences & Medicine, University of Surrey, Guildford GU2 7XH, UK; j.hunt@surrey.ac.uk (J.E.A.H.); a.mendis@surrey.ac.uk (J.R.M.); 3Centre for Perinatal Neuroscience, Department of Brain Sciences, Imperial College London, London SW7 2BX, UK; 4Intensive Care Unit, Ashford and St Peter’s Hospitals NHS Foundation Trust, Chertsey KT16 0PZ, UK; ian.white9@nhs.net (I.W.); jonathan.cooper@nhs.net (J.C.); karthik.somasundaram@nhs.net (K.S.); nik.shah@nhs.net (N.S.); 5Faculty of Sport, Allied Health & Performance Sciences, St Mary’s University, London TW1 4SX, UK; stephen.patterson@stmarys.ac.uk; 6William Harvey Research Institute, Barts and The London School of Medicine & Dentistry, Queen Mary University of London, London E1 4NS, UK; z.puthucheary@qmul.ac.uk; 7Institute for Sport, Exercise and Health, University College London, London W1T 7HA, UK; 8Centre for Human Health and Performance, Department of Medicine, University College London, London W1T 7HA, UK; h.montgomery@ucl.ac.uk; 9Intensive Care Unit, Royal Free London NHS Foundation Trust, London NW3 2QG, UK; 10Centre for Human and Applied Physiological Sciences, King’s College London, London WC2R 2LS, UK

**Keywords:** repetitive vascular occlusion stimulus, ICU-acquired weakness, blood flow restriction, critical illness, rehabilitation, muscle atrophy, vascular dysfunction

## Abstract

Muscle wasting is implicated in the pathogenesis of intensive care unit acquired weakness (ICU-AW), affecting 40% of patients and causing long-term physical disability. A repetitive vascular occlusion stimulus (RVOS) limits muscle atrophy in healthy and orthopaedic subjects, thus, we explored its application to ICU patients. Adult multi-organ failure patients received standard care +/− twice daily RVOS {4 cycles of 5 min tourniquet inflation to 50 mmHg supra-systolic blood pressure, and 5 min complete deflation} for 10 days. Serious adverse events (SAEs), tolerability, feasibility, acceptability, and exploratory outcomes of the rectus femoris cross-sectional area (RFCSA), echogenicity, clinical outcomes, and blood biomarkers were assessed. Only 12 of the intended 32 participants were recruited. RVOS sessions (76.1%) were delivered to five participants and two could not tolerate it. No SAEs occurred; 75% of participants and 82% of clinical staff strongly agreed or agreed that RVOS is an acceptable treatment. RFCSA fell significantly and echogenicity increased in controls (*n* = 5) and intervention subjects (*n* = 4). The intervention group was associated with less frequent acute kidney injury (AKI), a greater decrease in the total sequential organ failure assessment score (SOFA) score, and increased insulin-like growth factor-1 (IGF-1), and reduced syndecan-1, interleukin-4 (IL-4) and Tumor necrosis factor receptor type II (TNF-RII) levels. RVOS application appears safe and acceptable, but protocol modifications are required to improve tolerability and recruitment. There were signals of possible clinical benefit relating to RVOS application.

## 1. Introduction

More patients have been surviving admission to the intensive care unit (ICU) over the past decade [1,2,3]. These patients face a future of increased dependency and debility; 50% of working-age patients do not return to work and 70% require assistance with daily living activities in the year following discharge, with physical disability persisting for many years [4,5]. As a result, the UK’s National Institute of Health and Care Excellence (NICE) has declared post-ICU debility to be a public health issue [6].

ICU-acquired weakness (ICU-AW) affects approximately 40% of the adult ICU patient population [7], with a higher incidence rate (>60%) among patients who have had prolonged ventilation or severe sepsis [8,9]. It contributes substantially to functional limitation and impaired quality of life and is associated with poorer ICU outcomes (prolonged mechanical ventilation, increased rates of in-hospital and post-discharge mortality [10,11,12,13,14,15,16]) and, thus, increased healthcare costs, 18–30.5% higher hospitalisation costs, and further excess costs relating to the need for rehabilitation, frequent re-admissions, and social care upon discharge [11,13].

Whilst motor neuropathies can contribute to ICU-AW, skeletal muscle wasting resulting from an imbalance between protein synthesis and breakdown plays a dominant role in its pathogenesis [17]. In mechanically ventilated ICU patients, the cross-sectional area of the rectus femoris (thigh) muscle (RFCSA) decreases by 18% on average within 10 days of ICU admission, and more in those with multi-organ failure [18]. To date, no interventions are available to mitigate such muscle loss. Although some studies have shown beneficial effects, other studies have concluded there is insufficient evidence to support the efficacy of interventions, such as early mobilisation, physical activity [19,20,21,22,23,24], and non-volitional neuromuscular electrical stimulation (NMES) [25,26,27,28,29,30,31,32,33].

The repetitive application of a vascular occlusive stimulus (RVOS) might represent an effective mitigation strategy. RVOS involves the repeated inflation/deflation of a blood pressure cuff around a limb to above arterial pressure (~200 mmHg) to elicit brief bouts (~5 min) of limb ischaemia/reperfusion [34]. 

When applied before physical exercise, RVOS is associated with improvements in physical performance [35,36,37,38] and, when used in combination with low-intensity exercise (known as blood flow restriction exercise), with enhanced skeletal muscle hypertrophic and strength responses in athletes [39,40], healthy controls [41,42,43], and the elderly [44,45,46,47,48,49,50]. It seems to be effective in rehabilitation following a period of muscle unloading [51,52,53,54] and improves the physical function and health-related quality of life in patients with inflammatory muscle disease [55,56]. Furthermore, RVOS performed alone has been reported to mitigate atrophy and weakness induced by immobilisation and unloading in patients following surgery [34] or in healthy volunteers with experimentally induced limb immobilisation [57]. However, putative benefits in mitigating ICU-AW have yet to be explored. 

In addition, RVOS (alone or in combination with low-intensity exercise) is associated with improvements in local and systemic endothelial and microcirculatory function in healthy controls [58,59,60,61]. However, it is not known whether RVOS can mitigate the vascular dysfunction observed with bed rest or immobilisation [62,63,64,65] or critical illness [66,67,68,69,70,71]. Finally, RVOS may improve organ function at sites distant from its application (remote ischemic preconditioning) [60,72,73,74,75,76]. 

Thus, the application of RVOS to a single limb could potentially limit the degree of muscle wasting and vascular dysfunction observed in critically ill patients. We thus sought to determine the safety, tolerability, and feasibility of the RVOS application in such patients, and to obtain pilot data relating to impacts on skeletal muscle wasting, local vascular function, vascular biomarkers, and clinical outcomes including distant organ dysfunction.

## 2. Materials and Methods

### 2.1. Study Design

A pilot partially blinded interventional feasibility trial with randomisation was conducted. The study protocol has been published [77]. **Trial registration**: ISRCTN Registry, ISRCTN44340629. Registered on 26 October 2017.

### 2.2. Study Participants

Adult ICU patients from two UK hospitals (Royal Surrey County Hospital NHS Trust and Ashford and St. Peter’s Hospital NHS Trust) were recruited. A sample size of 32 participants was decided based on the recommended sample size for pilot and feasibility trials [78] and to allow for the balance in the stratification factors (gender and study site). Briefly, patients were eligible for recruitment if aged >18 years of age, within 48 h of ICU admission, receiving non-invasive (NIV/CPAP) or invasive mechanical ventilatory support, suffering a failure of >2 other organs (≥1 SOFA score on three domains including respiratory system), and likely to remain in the ICU for at least 4 days. Excluded were those with profound cardiovascular instability and coagulopathy and a history of or concurrent neuromuscular condition (neurological condition or muscle disease); peripheral arterial vascular disease history of deep vein thrombosis or pulmonary embolism; prior amputation of a lower limb; disseminated malignancy; or contraindication to pharmacological venous thromboembolism prophylaxis. The full exclusion criteria list is described elsewhere [77].

### 2.3. Study Protocol

All eligible patients at the time of consent were lacking the capacity to consent, hence the declaration of the agreement was sought from the patient’s ‘Personal Consultee’ who was a representative, partner, or close friend. Written retrospective informed consent was obtained once the participant recovered and was capable of informed consent. All participants received standard care according to local practice but were randomised in a 1:1 ratio to receive RVOS applied to their right proximal lower limb one session on day 1 and two sessions from day 2 to 10 of study enrolment or until ICU discharge (whichever occurred earlier). Each RVOS session included 4 cycles of 5 min cuff (12 × 124 cm SC12LTM segmental pressure cuff, Hokanson, WA, USA) inflation to 50 mmHg above the average systolic blood pressure (SBP) (average of three readings recorded over 3 h) for absolute arterial flow occlusion [79] followed by 5 min of complete deflation (0 mmHg) [34,57,59,60,80]. For participant safety, the maximum pressure used was 200 mmHg (even in cases with SBP of >150 mmHg). Control participants received no sham treatment. This study protocol was based on the RVOS protocol of previous studies assessing the effect of RVOS on muscle mass and strength during immobilisation and unloading [34,57].

Outcome measure assessments have been described in detail elsewhere [77]. Briefly, serious adverse events (SAEs) were assessed for safety, and tolerability was assessed using a pain visual analogue scale (VAS). Feasibility was evaluated against pre-specified criteria and acceptability through a semi-structured interview/ questionnaire. The ultrasound assessment of RFCSA and echogenicity (for muscle quality), vascular superficial femoral artery (SFA) diameter, and blood velocity and flow at rest and following 5 min of ischemia (to assess flow-mediated dilation (FMD) and reactive hyperaemic response) were evaluated at day 1, 6, and 11 of study enrolment or until ICU discharge if earlier than day 11. All ultrasound scans were performed by one person and good intra-observer reproducibility was observed in RFCSA (healthy controls *n* = 11, correlation coefficient 0.98, 95% CI 0.90–0.99, and ICU patients *n* = 6, 0.99, 95% CI 0.93–0.99), resting SFA diameter (0.96, 95% CI 0.88–0.99), and blood flow (0.87, 95% CI 0.59–0.96) in healthy controls (*n* = 13).

Dominant handgrip strength (Takei digital dynamometer) and overall muscle strength (Medical Research Council Sum Score, MRC-SS) were assessed if off sedation and CAM-ICU negative on days 6 and 11 and at ICU and hospital discharge. Physical function was assessed using the ICU mobility score [81] on days 1, 6, and 11 of study enrolment or until ICU discharge if earlier than day 11, and ‘timed up and go’ (TUG) and ‘sit to stand’ (STS) were assessed at hospital discharge. The association of RVOS intervention with clinical outcomes. The association of RVOS intervention with clinical outcomes including delirium; acute kidney injury (urine output assessed using the AKIN classification [82]; organ support, length of hospital stay, and mortality were assessed. Muscle metabolites, vascular adhesion molecules, and growth factors as well as inflammatory cytokines were measured using commercially available Quantikine ELISA and human magnetic multiplex Luminex assays (R&D Systems, Minneapolis, MN, intra-assay CV <6%).

### 2.4. Statistical Analysis

Statistical analysis was conducted using GraphPad Prism 9.0.0; the Shapiro Wilks normality test was used to check for normality, and the level of statistical significance was set at alpha 0.05. Baseline characteristics were presented as the mean (standard deviations, SD) for parametric data and median (interquartile range, IQR) for non-parametric data. Descriptive statistics were summarised as frequencies (%) and proportions or as free text. Categorical variables were displayed as frequencies (%) and continuous variables were reported as the mean ± standard error of the mean (SEM) for normally distributed data and median (interquartile range) for not normally distributed data. A comparative two-way ANOVA, or mixed effects model analysis if missing values with post hoc Bonferroni’s multiple comparisons test for repeated measure parametric data, was applied to determine change over time, but was not reported on exploratory/outcome measures between group comparisons as the study was not suitably powered for such analyses.

## 3. Results

Eligible multi-organ failure admitted ICU patients were recruited over 16 months and participant flow through the study is illustrated in the consort diagram (Figure 1). A total of 12 (54.5%) agreed to enrolment and were randomised to the control (*n* = 6) and intervention group (*n* = 6). Baseline characteristics of treatment groups were similar (unpaired *t*-test, *p*-values > 0.05) (Table 1).

### 3.1. Feasibility

The success rate of trial processes was compared against pre-specified criteria (Table 2).

The table presents the study results achieved for trial processes against set pre-specified feasibility targets.

Screening, consent, and recruitment: The majority of admitted patients (85.8%) were ineligible for participation (Appendix A). Of those approached, only 54.5 % agreed to enrolment (less than the >75% sought), and only 12 of the intended 32 patients (38%) were recruited over 16 months.

Randomisation procedures: These worked well, with a balance in demographic data and illness severity between the groups (Table 1).

*Implementation of RVOS:* In total, 76.1% (67 of 88) of scheduled RVOS sessions were performed, slightly lower than the pre-specified feasibility target of 80%. On eight occasions, the average SBP of the participants was >150 mmHg prior to commencing RVOS, however, to maintain participant safety, a maximum cuff pressure of 200 mmHg was used instead of 50 mmHg above SBP. Reasons for failure to deliver the rest of the RVOS sessions are listed in Table 2. Only 45.6% (*n* = 5) of participants remained in the ICU for the full 10 days of study enrolment, less than the >50% retention rate sought.

Data collection/outcome assessment: All possible RFCSA ultrasound scans were acquired; 80.8% of scheduled SFA vascular outcomes and >90% of strength and functional outcome measures were obtained. Reasons for the failure to perform the remaining vascular, strength, and functional outcome measures are listed in (Appendix A).

Serious Adverse Events: No SAEs related to study participation occurred. SAEs unrelated to study participation included one left leg DVT and two deaths (one during the ICU stay and one during a hospital stay) in control subjects; in intervention subjects, there was one de novo diagnosis of myotonic dystrophy (participant excluded from analysis) and two deaths (one during the ICU stay and one during a hospital stay). One control participant reported numbness on the right lateral thigh following SFA outcome measures which involved cuff inflation to 200 mmHg for 5 min. This was diagnosed as a right lateral cutaneous nerve of the thigh palsy and required no medical treatment.

### 3.2. Tolerability of RVOS

Tolerability assessment could not be performed following delivery of >70% RVOS sessions due to participants being sedated to facilitate mechanical ventilation (Appendix A). At least one VAS score was obtained from four out of six intervention participants and two subjects rated RVOS with a pain score >8 out of 10. Two participants could not tolerate the standard protocol of a cuff pressure 50 mmHg above SBP, and the cuff thus had to be deflated to a level that was acceptable for the individual (95 mmHg and 120 mmHg), however, the acceptable cuff pressures were below the subject’s SBP (average SBP was 127 mmHg and 144 mmHg). 

### 3.3. Acceptability

A semi-structured interview with participants (*n* = 7) and their personal consultees (*n* = 2) was conducted to assess trial acceptability at hospital discharge. All assessed reported satisfaction with the conduct of the study and suggested no protocol improvements. All agreed it was reasonable to approach personal consultees when patients were not capable of providing consent for themselves. One intervention participant found the standard RVOS protocol (cuff pressure 50 mmHg above SBP) painful, but acceptable when the cuff pressure was reduced to below the SBP. Three out of four (75%) intervention participants strongly agreed that RVOS would be an acceptable treatment if found to be effective. One participant could not recall the intervention procedure and, therefore, did not provide a response. Personal consultees thought that RVOS was “non-invasive” and should be used “if patients don’t find it painful”. 

Eleven clinical staff members (including consultants, nurses, physiotherapists, and dieticians) felt that RVOS did not affect their clinical practice, with one commenting, “this is an excellent intervention routinely used in the rehabilitation programme and it will be interesting to see how it translates to the critically ill patients”. Three (27.3%, all physiotherapists) clinical staff reported that RVOS sessions disturbed their normal routine. Four (36.4%) clinical staff strongly agreed, five (45.5%) agreed, and two (18.2%) were neutral to the statement, “If the intervention was found to be effective at reducing muscle weakness, it is an acceptable treatment”.

### 3.4. Exploratory Measures

Although not powered to detect the effectiveness of the RVOS intervention, the effects of RVOS on muscle mass, vascular function, clinical outcomes, and blood biomarkers were explored to provide direction for future clinical trials. A consort diagram with the sample size for outcome measures at each time point of the study is available in Appendix A.

#### 3.4.1. Effect on Rectus Femoris Cross-Sectional Area and Echogenicity

Representative day 1 and 6 RFCSA ultrasound images are shown in Figure 2. Due to the small sample size on day 11 (two participants per treatment group, because of either ICU discharge or death), this time point was excluded from the analysis. Furthermore, a comparison between groups and between the treated and untreated legs of intervention participants was not performed due to the small sample size. RFCSA fell significantly between day 1 and day 6 in both control and intervention subjects (two-way ANOVA with Bonferroni’s multiple comparisons, time *p* < 0.05 *). The Median (interquartile) percentage RFCSA decrease on day 6 was −14.6% (−26.3 to −4.3%) in the controls and −17.4% (−18.5 to −4.4%) in the intervention group (Appendix A).

The impact of RVOS on muscle quality was assessed by measuring RF echogenicity. Subcutaneous fat tissue can affect ultrasonography imaging quality and thus echogenicity. The ultrasound attenuation is higher (and the image thus darker and echogenicity values lower) in muscle examined at greater depth and, thus, should be corrected for subcutaneous fat thickness (SFT) and can be calculated as uncorrected echogenicity + (subcutaneous fat thickness [cm]) × 40.5278) [83,84]. SFT corrected RF echogenicity increased significantly between day 1 and day 6 in all participants (two-way ANOVA with Bonferroni’s multiple comparisons, time *p* < 0.05 *). The median (interquartile) percentage increase in SFT corrected RF echogenicity was 6.8% (−0.4 to 23.4%) and 22.1% (13.5 to 29.7%) in the control and intervention groups, respectively, on day 6. Uncorrected echogenicity showed a similar but non-significant trend (Appendix A).

The RFCSA of the treated right and untreated left leg of intervention participants were compared to assess RVOS effects on local and remote tissue. Ultrasound images of one participant were excluded due to poor quality. In both the untreated and treated legs, a decrease in RFCSA and increase in SFT corrected echogenicity was statistically significant (two-way ANOVA with Bonferroni’s multiple comparisons, time *p* < 0.01 **). The median percentage decrease in RFCSA and increase in SFT corrected echogenicity in both the untreated and treated legs are presented in Appendix A.

#### 3.4.2. Effects of RVOS on Muscle Function

Five out of seven participants (71.4%) scored < 48 in the MRC-SS muscle strength test during their ICU stay, and, therefore, had ICU-AW [85]. Day 11 of study enrolment (only one participant was assessed on day 11) was excluded from repeated measures statistical analysis. The MRC-SS score of both control and intervention participants improved during the hospital stay but did not reach statistical significance (mixed effect model analysis with post hoc Bonferroni’s multiple comparison test, *p* = 0.051). At ICU discharge, the median (interquartile range) MRC-SS score was 53, 50–56 in intervention subjects and 46, 36–48 in controls (Appendix A).

An objective measure of strength using a dominant handgrip strength assessment confirmed greater weakness in ICU-AW patients; median (interquartile) handgrip strength was 9.6 kg (5.1–13.8 kg) and 21.15 kg (19.2–23.1 kg) in ICU-AW (*n* = 5) and non-ICU-AW patients (*n*= 2), respectively. There was no significant change in hand grip strength over time (*p* > 0.05) in control and intervention subjects. At ICU discharge, the median handgrip strength of intervention participants was 10.2 kg (7.2–11.8 kg) in the control and 14.1 kg (12.3–15.8 kg) in the intervention group (Appendix A).

Because all participants were sedated at enrolment, they had no active movement and scored 0 on the ICU mobility scale. The same applied to the three participants remaining in ICU on day 11, and this time-point was excluded from repeated measure statistical analysis. Mobility scores significantly improved during the hospital stay in both control and intervention participants (mixed effect model analysis with Bonferroni’s multiple comparisons, *p* < 0.01 **). At ICU discharge, intervention participants had mobility (median (interquartile range) scores of 8 (8–8: i.e., able to walk with the assistance of one person) and 5 (3–7: i.e., able to step or shuffle transfer from bed to chair) in controls (Appendix A. The median (interquartile range) STS performed within 30 s was 8 (4–11) in the controls and 7 (0–7) in the intervention group, and the time taken to perform the 3 metre TUG was 19 (8–34) seconds and 15 (0–20) seconds in the control and intervention groups, respectively (Appendix A).

#### 3.4.3. Effects of RVOS on Vascular Health

No significant differences were observed in resting diameter, blood velocity, and blood flow over time in both controls and intervention participants (mixed effect model analysis with Bonferroni’s multiple comparisons, *p* > 0.05). At enrolment, the mean (SEM) resting SFA diameter was 5.6 (0.4) mm in controls (*n* = 5) and 6.5 (0.9) mm in intervention participants (*n* = 3); by day 6, SFA mean diameter in controls decreased by −1.8%, while it increased by 6.2% in the intervention group. 

To assess the reactive hyperaemic response and FMD, peak SFA diameter, and blood velocity and flow were measured immediately following 5 min of vascular occlusion. Peak SFA diameter, FMD, blood flow, and velocity did not change significantly by day 6 in either control or intervention groups (Appendix A). However, similar to resting diameter, by day 6, the peak mean SFA diameter in controls decreased by −3.4%, while it increased by 5.3% in the intervention group.

#### 3.4.4. Effects of RVOS on Clinical Outcomes

For control and intervention subjects, respectively, during the first 11 days (or until ICU discharge or withdrawal), median (interquartile range) days of invasive mechanical ventilation before successful extubation were 5 (4–30) days (*n* = 5) and 3 (1–27) days (*n* = 4); duration of noradrenaline use were 4 (2–8) days (*n* = 5) and 2 (1–7) days (*n* = 5); 3/4 (75%) controls and 4/4 (100%) intervention subjects had incidence of delirium, and the median (interquartile range) percentage of awake days in which delirium was detected were 67% (20–88%) (*n* = 3) and 33% (33–58%) (*n* = 4). 

At enrolment, the mean (SD) serum creatinine concentration was 157 (51) µmol/L in controls and 127 (73) µmol/L in intervention subjects. Three participants in each treatment group had AKI at ICU admission; no further intervention subjects developed AKI according to urine output of AKIN criteria [80]; compared to all five controls, the median (interquartile range) days with AKI was 3 (1–5) days during the first 11 days of study enrolment or until ICU discharge (if occurred before day 11). The median (interquartile) days participants received any renal replacement therapy (RRT) was 3 (1–4) days in control (*n* = 5) and 0 (0–2) days in the intervention subjects (*n* = 5), respectively. 

The mean (SD) total SOFA score at enrolment was 12 (2) in control and 10 (3) in intervention subjects. During the 11 days of study enrolment or until ICU discharge (if before day 11), the total SOFA score reduced significantly in both treatment groups (*p* < 0.05), with the mean (SEM) percentage decreasing by 36% (11) and 46% (10) in controls and intervention participants, respectively. 

The mean (SEM) length of ICU stay was 20 (8) days in controls and 16 (9) days in intervention participants; the median (interquartile range) hospital stay following ICU discharge was 5 (2–30) and 11 (3–19) days in controls and intervention, respectively; and the 28-day ventilator-free survival was 23 (7–24) days in controls and 24 (0–26) days in the intervention group. The in-hospital mortality rate was 33% (2/6 participants) in controls and 40% (2/5 participants) among intervention participants. No further mortality was reported 3 months following hospital discharge.

#### 3.4.5. Effects of RVOS on Blood Biomarkers

The mean (SEM) IGF-1 levels increased by 59.8% on day 6 (87.1 (12.0) ng/mL) and remained high at hospital discharge (84.5 (7.6) ng/mL) in the intervention group, while the levels were similar throughout the study enrolment in the controls (Figure 3A). Circulatory syndecan-1 levels were higher in the acute phase of the illness, its mean (SEM) levels in the control group increased by 19.6% (8722.6 (522.3.9) pg/mL) and 6.1% in the intervention group (6173.1 (497.4) pg/mL) at day 6; the levels reduced in both groups at hospital discharge (Figure 3B). Similarly, IL-4 and TNF-RII levels were high at enrolment and the levels remained high in the control group (IL-4 59.5 (6.5) pg/mL and TNF-RII 15390.6 (1847.1) pg/mL) at day 6, while in the intervention group, IL-4 was reduced by 11.6% (36.6 (1.6) pg/mL) and TNF-RII by 16.6% (11879.4 (935.2 pg/mL)(Figure 3C,D). Other blood biomarkers such as muscle metabolites, vascular adhesion molecules and growth factors, and inflammatory cytokines were also assessed and the results are presented in Appendix A. 

## 4. Discussion

This was the first study to assess the safety, tolerability, feasibility, and acceptability of RVOS application to patients with multi-organ failure, and to explore the potential to mitigate the development of ICU-AW. Significant barriers to recruitment were observed, such as fewer eligible patients and a higher proportion of personal representatives declining agreement to enrolment in the study, which would need to be addressed in the design of a future large RCT.

The study was unable to deliver the intervention at the intended frequency and intensity due to participants declining to receive RVOS sessions and intolerability to a cuff pressure of 50 mmHg above SBP.

Outcome measures were successfully collected, although a review of the timeframe of intervention delivery and serial outcome measures (fewer data collection points in ICU) is required. This study recommends changes to the methodology to avoid excessive attrition due to intolerance to procedure (vascular measures) and intervention.

### 4.1. Primary Outcome Measures

Multi-organ failure patients undergo more severe muscle loss compared to single-organ failure patients [18] and were, therefore, chosen as study participants as they would benefit most from an intervention that mitigates ICU-AW development. However, around 85% of screened patients were ineligible for recruitment due to profound coagulopathy or cardiovascular instability, often with poor prognosis, and additional co-morbidities such as malignancy, neuromuscular condition, peripheral arterial vascular disease, and a history of DVT or PE. 

Enrolment in the early stage of illness was important as muscle wasting occurs rapidly within days of critical illness [18] and an early application of intervention would be more fruitful than at later stages of illness when a large proportion of the muscle wasting would have already occurred. However, the limited recruitment window occuring within 48 h of ICU admission and hospitalisation <48 h prior to ICU admission was a barrier to recruitment. A lower agreement rate for enrolment and low eligibility meant the recruitment target was not achieved.

Widening the inclusion and reducing the exclusion criteria might improve recruitment rates. Specifically, we excluded patients with a history of peripheral arterial vascular disease, although RVOS appears safe in such patients [86,87,88]. Another potential strategy to improve recruitment in a future RCT is to set up the study at multiple (>2) sites, although the protocol is highly demanding and participating centres might be unable to deliver the interventions and assessments per protocol.

Seeking agreement from personal consultees limited recruitment, a higher proportion of personal representatives tend to decline study enrolment than patients [89]. Reasons for declining participation were not collected but reasons may include uncertainty regarding the patient’s wishes or an intuition that the patient would not want to participate or would be too upset to contemplate participation [90]. However, there is no ethically acceptable alternative to this. 

All RFCSA measurements were performed; however, around 20% of SFA measures were missed and some were due to intolerance. In this study, SFA was examined instead of typically assessed brachial artery to avoid common wrist and arm arterial lines in ICU patients, and our interest to assess the changes in the regional vascular function because of local RVOS application. However, participants might find brachial FMD to be more tolerable due to the smaller area of tissue being under ischemia compared to SFA FMD and it would be an indicator of systemic changes in endothelial function.

A total of 67 RVOS sessions were delivered to five participants and no serious adverse events occurred in relation to participation, although one control participant suffered numbness of the lateral thigh following assessment of arterial outcome measures. An SC12LTM segmental pressure wide cuff (Hokanson, WA, USA, 12 × 124 cm) was used in this study such that pressure was dispersed over a greater contact area, thus reducing the risk of underlying soft tissue injury and aiding blood flow occlusion at lower pressures [91,92]. However wider cuffs have been reported to affect nerve conduction more than narrow cuffs, therefore, care must be taken during cuff application [93]. To minimise risk, it is preferable to determine and apply the lowest cuff pressure (arterial occlusion pressure) required for blood flow occlusion; modern cuffs (such as the automatic personalised tourniquet system; Delfi Medical, Vancouver, BC, Canada) can automatically do this [54,94]. Arterial occlusion pressure considers factors such as cuff shape and width and limb characteristics in addition to SBP [95]. Previously, the average arterial occlusion pressure recorded with a wide cuff was lower than 50 mm Hg above SBP in both healthy controls [96] and surgical patients [97]. 

More than 50% (*n* = 11) of the missed sessions were due to the participant declining the session. This study protocol involved the application of RVOS sessions twice a day (>4 h apart) which increased the burden of study participation. Critically ill patients commonly suffer from fatigue [98] and low mood [99], which could have contributed to such lack of engagement. Therefore, perhaps RVOS can only be applied when patients are sedated and not tired by subsequent physical rehabilitation efforts.

In addition, almost 1/5 of missed RVOS sessions (*n* = 4) were terminated due to intolerability, with two out of four participants rating RVOS with a pain score of >8 out of 10 on the VAS scale (0 being no pain and 10 being worst possible, unbearable, and excruciating pain). However, tolerability data obtained in the study was limited as participants remained sedated during 70% of RVOS session delivery. When participants did not tolerate the cuff pressure of 50 mmHg above SBP, a lower cuff pressure that was acceptable for the individual (120 and 95 mmHg) was used, however, the acceptable cuff pressures were not above the individual’s SBP and therefore might have a reduced protective stimulus. The “optimal pressure” required for inducing the protective effects of RVOS is unknown. Although evidence that lower cuff pressure is fully effective is very limited, previously, a cuff pressure as low as 50 mmHg has been reported to reduce the unloading induced muscle weakness [100]. Furthermore, recently, the combination of NMES and blood flow restriction with a cuff pressure of 40–80% of arterial occlusion pressure has been investigated which could possibly be more tolerable due to lower cuff pressure [101,102,103]. Studies that applied RVOS to relatively young, healthy individuals did not report any tolerability issues [34,57,104]; however, amongst patients with cardiovascular risk factors undergoing hip fracture surgery, 2% (8/316) failed to complete RVOS sessions due to discomfort [105]. Tolerability might be improved if the applied pressure progressively increased with each session until the participant became familiar with the sensation. Furthermore, modern automated personalised tourniquet systems (above) might be better tolerated. 

The pre-specified retention rate of >50% was not achieved. It is notoriously difficult to accurately predict the duration of ICU stays at the time of admission. RVOS sessions were applied twice daily for 10 days, based on a previous study showing benefit for 11 days of intervention following knee surgery [34]. To date, other studies have reported that the application of one to two daily sessions of RVOS for 1 to 8 weeks results in reduced disuse atrophy and weakness, and in improved vascular function [34,57,59,60]. However, no studies have aimed to determine the optimum frequency and length of RVOS application required for its protective effects and such studies should be conducted in the future.

RVOS was described as a “non-invasive” intervention and was acceptable among participants and their personal consultees and clinical staff. Three of four (75%) intervention participants strongly agreed, while 82% (9/11) of clinical staff strongly agreed or agreed that RVOS is an acceptable treatment if it is found to be effective. 

Overall, this study showed that RVOS application may be safe and acceptable, but barriers to recruitment meant we were unable to demonstrate the feasibility of study conduct with the current protocol.

### 4.2. Exploratory Measures

Both the control and intervention participants exhibited significant muscle loss following enrolment on a scale comparable with past reports [18]. Given the limited sample size, this study was not powered to identify the effectiveness of RVOS in mitigating muscle atrophy in ICU patients. Daily application of RVOS reduced disuse knee extensor muscle atrophy over the 11 days following surgical ligament reconstruction in relatively healthy and young (controls aged 23.0 ± 2.5 years, intervention aged 22.4 ± 2.1 years) individuals [34]. Our study population was, of course, older and sicker, and disuse is only one element driving muscle wasting in ICU [106]. Moreover, studies that report effective mitigation of disuse muscle atrophy delivered two sessions for 10–14 days unlike the median (IQR) 10 (5–19) sessions achieved in this study. Routine administration of propofol may also impede the cardio-protective effect of remote ischemic preconditioning (akin to RVOS) [107]. Future studies should take this into account. 

SFT-corrected RF echogenicity significantly increased on day 6 in both groups (*p* < 0.05), consistent with previous reports on the critically ill [108,109,110]. Echogenicity is inversely related to muscle quality [110,111,112,113], with increased echogenicity resulting from the replacement of muscle with fat [114] or fibrous tissue [112], or muscle necrosis [113]. Uncorrected echogenicity also increased; however, the percentage changes did not reach statistical significance unlike SFT corrected echogenicity, and this suggests researchers should consider the SFT to confidently evaluate echogenicity in the future.

### 4.3. Muscle Function

Five of the seven participants (71.4%) had ICU-AW defined as MRC-SS of <48 [85]. This incidence rate was higher than previously reported [8,9,14,115]. The profound muscle wasting in multi-organ failure patients compared to single organ failure patients likely accounts for the higher ICU-AW incidence rate in our study cohort. The objective evaluation of handgrip strength showed participants who met the criteria for ICU-AW according to MRC-SS had lower hand grip strength than non-ICU-AW during their ICU stay, supporting the validity of this measure. In addition, hand-held dynamometers could have been used to assess the lower limb muscle strength local to the application of RVOS. Higher strength (MRC-SS score and handgrip strength) and physical function (ICU mobility score) at ICU discharge was observed in the intervention group, however, this study was not powered for the comparison of strength and physical function between treatment groups. Moreover, the mobility achieved could have been influenced by pre-ICU factors such as age, comorbidities, and physical function before the critical illness. Previous investigations administering RVOS during periods of unloading have reported protection against loss in muscle strength, albeit in healthy controls [55,98]. 

### 4.4. Vascular Measures

The resting SFA diameter was similar to that previously reported [116]. Values did not change over 6 days in ICU, although an intervention-related trend for structural enlargement was compatible with the vascular remodelling observed in blood flow restricted exercise [117]. Future studies should measure the maximal dilatory capacity for ischemic exercise or sublingual administration of pharmacological vasodilators (such as glyceryl trinitrate) to confirm the structural enlargement [118]. Lower resting and peak blood velocity observed in the intervention group compared to controls could be explained by a larger SFA diameter, as the velocity varies inversely with the total cross-sectional area of a blood vessel [119]. There was a trend of lower peak blood velocity and blood flow during reactive hyperaemia on day 6 in both treatment groups, which is an indicator of microvascular dysfunction. The lack of significant change in the FMD could have been due to the potential structural changes to the artery as the increase in artery diameter decreased the shear stress stimulus, resulting in a small dilatory response [120]. All participants at baseline and one participant per group on day 6 were receiving the vasopressor noradrenaline via infusion as part of their treatment, which could have possibly influenced our results, although vasopressors may not in fact affect endothelial function [121]. 

Short (7 days, daily 4 × 5 min) or long-term (8 weeks, 3 times a week) RVOS has been reported to improve endothelial function in healthy young individuals [59,60]. However, endothelial function declines with age [122], as may related RVOS impacts [123]. 

The effectiveness of RVOS could also differ between vessels, as endothelial function is better preserved in upper than lower limb arteries [124] and endothelial dysfunction is more pronounced in lower limb arteries among patients with arterial diseases [125]. Hence, the effectiveness of reversing the endothelial dysfunction might vary depending on the vessel and the extent of endothelial dysfunction due to ageing and illness [71].

### 4.5. Clinical Outcome

RVOS might be a protective effect against remote target organs. Lower rates of AKI incidence (3/5 vs. 5/5) (according to urine output criteria of AKIN classification [82] and a greater decrease in the SOFA score (46 (10) vs. −36(11) %) were observed during the first 11 days of study enrolment or until ICU discharge in the intervention group. However, the small size reduces the validity of this finding and may prevent extrapolation. Nevertheless, in agreement, previous studies of cardiac surgical patients have reported RVOS to be associated with lower rates of AKI, or renal replacement therapy, and with enhanced renal recovery [73,75,126,127,128]. Hence, this protective effect should be explored in bigger RCTs. The exact mechanism by which RVOS confers this protective effect is unclear. One potential mechanism includes endogenous mediators that could activate the signalling molecules such as protein kinase C, which, via signal transduction pathways, lead to the opening of the mitochondrial KATP channels inducing protective effects such as reduced apoptosis and improved cell survival [127,129,130].

The incidence of death in our study cohort was comparable to that previously reported in a similar patient cohort [131]. No effect of RVOS on mortality has been reported previously, however, reduced ICU stay was observed in cardiac surgical patients [75].

### 4.6. Blood Biomarkers

A catabolic state during critical illness decreases the growth factors/IGF-1 axis. The resulting lower circulatory IGF-1 levels [132,133] are linked to the critical illness severity and poor prognosis [14,134], with a sustained reduction in patients with muscle wasting [135]. The trend of elevated IGF-1 levels in the intervention group could mitigate muscle wasting via upregulation of the IGF-1/Akt/mTORC1 anabolic signalling pathway [136]. Previously, an acute increase in circulatory IGF-1 levels within 15 min of a single and after 2 weeks of blood flow restriction exercise training has been observed in healthy young men [137,138]. A similar loss of RFCSA occurred between the treatment groups, perhaps suggesting that observed elevated IGF-1 levels did not stimulate muscle protein synthesis. This could be possibly due to the levels of IGF-1 not reaching the required levels [138,139] or an intrinsic secretion of muscle IGF-1 might be a key determinant for switching on anabolic pathways [140,141,142,143].

Syndecan-1, an integral membrane protein with antiadhesive and anticoagulant properties, protects the endothelium and maintains the vascular barrier function [144]. Critical illness increases circulatory syndecan-1 levels and is associated with increased mortality [145]. A significant change in syndecan-1 levels was observed, with high levels in the early phase of the critical illness. The increased circulatory syndecan-1 levels correlate with muscle echogenicity changes [146]. Moreover, it is inversely associated with FMD in nephrotic syndrome patients [147]. A smaller increase in the circulatory syndecan-1 observed on day 6 in the intervention participants could suggest RVOS potentially prevents the shedding of syndecan-1 and maintains endothelial integrity and its function. 

The trend of reduced IL-4 levels was observed in the intervention group on day 6. Previously, no significant change in IL-4 levels was reported within an hour of a single RVOS session in the healthy controls and patients undergoing elective surgery [148,149], while the levels were upregulated in the animal models at 48 h [150]. The observed contrasting results could be due to either species variation, acute or multiple application of RVOS, or the length of interval after the intervention IL-4 levels were measured. In-vitro human endothelial cell culture model studies have shown that IL-4 induces oxidative stress and inflammation in the vascular endothelium [151,152], therefore the observed trend of lower IL-4 levels in the intervention group could be beneficial. The lack of improvement in the endothelial function of local SFA (based on the FMD data), despite the reduction in the vascular dysfunction markers syndecan-1 and IL-4, could potentially be due to the conduit artery structural enlargement. Previous studies showed exercise training-induced initial increase in conduit artery endothelial function was superseded by arterial structural remodelling [153,154]. The increase in vascular structure normalises the shear rate levels which results in NO-mediated endothelial function returning to the initial levels.

TNF-α mediates its signalling via the activation of two TNF-α receptors; TNF-RI, expressed in all cell types, and TNF-RII, found mainly in the leukocytes and the heart [155]. TNF-RII levels reduced significantly over time during the hospital stay. TNF-α activates the endothelial cells and leukocytes which leads to inflammatory responses including leukocyte rolling, firm adhesion between endothelial cells and leukocytes, and transmigration of leukocytes from blood to tissue. TNF-RII is essential for TNF-α-induced expression of adhesion molecules such as VCAM-1 and ICAM-1 and its absence results in the impairment of endothelial cells and leukocytes interaction, which is a critical step in the inflammatory response [156]. Therefore, a decrease in the circulatory TNF-RII levels could potentially suggest a reduction in inflammation, which could be beneficial in mitigating inflammation-induced muscle atrophy.

### 4.7. Limitations

The small sample size limits the inferences on safety and efficacy, and outcome results are hypothesis-generating. The nature of the intervention and lack of sham treatment in the control group meant that participants and researchers were not blind to treatment allocation. However, muscle and arterial ultrasound images were analysed blinded to subject identity, study time point, and group allocation.

The control group received 5 min of cuff inflation to 200 mmHg, followed by reperfusion (ischemic reperfusion stimulus) at a maximum of 3 occasions during arterial vascular measures assessment. However, this ischemic reperfusion stimulus is unlikely to have had a major biological impact given the brevity (5 min) and infrequency (one cycle on days 1, 6, and 11 of study enrolment) of the stimulus. The previous study reports that multiple cycles of brief ischemia are required to meet the ‘ischemic threshold’ and elicit a protective effect [157].

Because the majority of RVOS sessions (>70%) were delivered when participants were mechanically ventilated and sedated, only limited tolerability data were collected, and these could have been influenced by delirium which is common in critically ill patients. Volitional assessments such as hand grip strength and MRC-SS could not be performed on sedated patients. In addition, the >50% of participants discharged from ICU by day 10 could not receive all possible RVOS sessions or day 11 muscle and arterial ultrasound and strength measures assessments. We could not control for variable intensity and duration of use of the vasopressors, corticosteroids, sedation, neuromuscular blocking drugs, or physical therapy in such a small sample.

## 5. Conclusions

The application of RVOS to ICU patients appears safe and acceptable, although not readily tolerated by all patients. The potential remote protective effect of RVOS on renal function and beneficial impacts on strength and mobility should be studied in future multicentre RCTs. Recruitment and retention of ICU patients with multi-organ failure were difficult, and protocol changes such as widening the inclusion criteria, increasing the recruitment sites, and altering the frequency of intervention and outcomes measures, might enhance the ability to perform such studies. RVOS induced upregulation in growth factors IGF-1 and reduced vascular dysfunction markers syndecan-1 and IL-4 and inflammatory marker TNF-RII which may offer protection against muscle wasting, vascular dysfunction, and inflammation. However, the small sample size reduces the validity of these findings which should be confirmed in future studies.

## Figures and Tables

**Figure 1 jcm-11-03938-f001:**
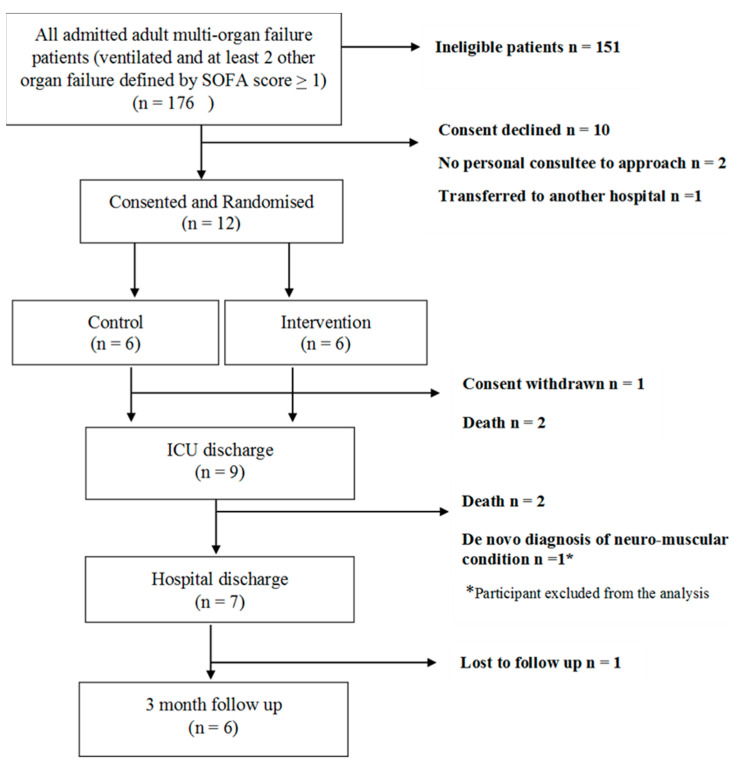
Study consort diagram.

**Figure 2 jcm-11-03938-f002:**
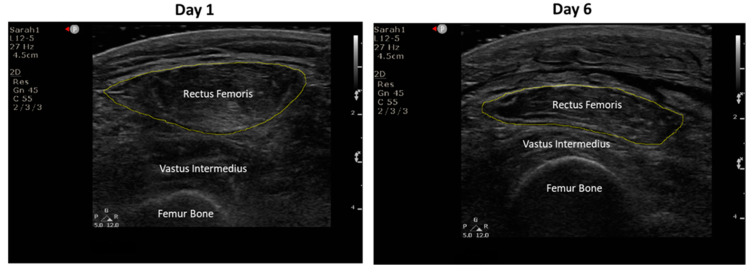
Representative ultrasound images of Rectus femoris cross-sectional area (RFCSA) on days 1 and 6 of study enrolment.

**Figure 3 jcm-11-03938-f003:**
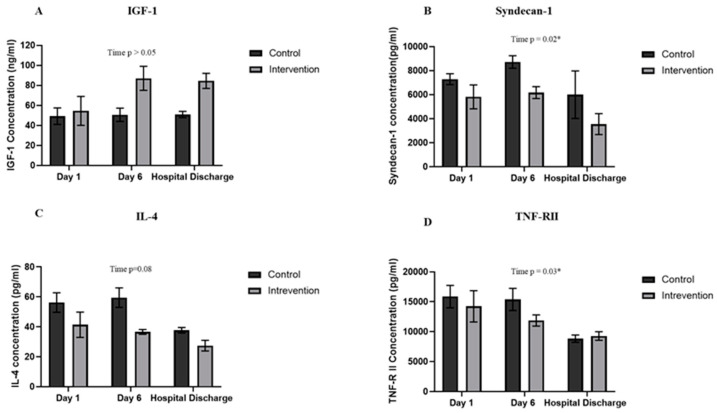
Graphs show the mean (SEM) circulatory levels of IGF-1 (**A**), Syndecan-1 (**B**), IL-4 (**C**), and TNF-R II (**D**) in the control and intervention groups on days 1 and 6 of study enrolment (control *n* = 5, intervention *n* = 4) and at hospital discharge (control *n* = 2, intervention *n* = 3). Repeated measure analysis of the mixed effect model with post hoc Bonferroni’s multiple comparison test was used. Symbol * represents a *p*-value < 0.05.

**Table 1 jcm-11-03938-t001:** Baseline characteristics of the control and intervention participants.

Baseline Characteristics	Control (*n* = 6)	Intervention (*n* = 6)
Age, mean (SD) (years)	65 (10)	70 (11)
Gender, male/female (no)	4/2	4/2
BMI, mean (SD) (kg/m2)	29.8(5.0)	29.2(6.8)
Charlson Co-morbidity Index, median (IQR)	3 (1–5)	3 (2–5)
APACHE II Score, mean (SD)	22(6.4)	19.5(7.3)
ICNARC Score, mean (SD)	34(9.7)	24(4.7)
Katz Index, median (IQR)	6(6–6)	6(6–6)
MUST score, median (IQR)	2 (2–2)	2 (2–3)
Hospital length of stay prior to ICU admission, median (IQR) (days)	0 (0–1)	0 (0–1)
SOFA score at ICU admission, mean (SD)	12 (1.2)	10 (2.6)
Primary Diagnosis, No (%)		
CAP	2 (33.3)	3 (50)
Pulmonary Oedema	1 (16.7)	
Ischemic Bowel	2 (33.3)	1 (16.7)
Septic Shock	1 (16.7)	
Acute Pancreatitis		1 (16.7)
AKI	3 (50)	3 (50)
Comorbidities, No (%)		
Asthma	2 (33.3)	
Chronic Kidney Disease	1 (16.7)	
COPD	1 (16.7)	1 (16.7)
Crohn’s Disease		1 (16.7)
Diabetes Mellitus (Type I and Type II)	1 (16.7)	2 (33.3)
Hypertension	1 (16.7)	3 (50)
Previous Cerebrovascular accident	1 (16.7)	
Osteoarthritis	1 (16.7)	1 (16.7)

AKI—acute kidney injury, APACHE II—acute physiology and chronic health evaluation II, BMI—body mass index, CAP community acquired pneumonia, COPD—chronic obstructive pulmonary disease, ICNARC intensive care national audit and research centre, IQR interquartile range, MUST—malnutrition universal screening tool (identifies adults at risk of malnutrition; Score 1 Low Risk; Score 1 Medium Risk; Score 2–6 High Risk), and SOFA—sequential organ failure assessment safety.

**Table 2 jcm-11-03938-t002:** Feasibility assessment of trial process in comparison to pre-specified criteria.

Trial Process	Feasibility Target	Achieved	Comment
Screening	<55 of potentially eligible patients being missed	176/176 (100%) screened	
Consent	>75% agreement	54.5 %	
Recruitment Rate	32 patients within 16 months	12 patients were recruited within 16 months	
Randomisation	Balanced demographic and severity of illness in intervention and control arm	Groups were balanced (Table 1).	
Delivery of Intervention	80% of the scheduled RVOS sessions performed	76.1% (*n* = 67/88)	Rest (*n* = 21) not delivered due to participants declining (12.5%, *n* = 11); intolerance (4.5%, *n* = 4); participant unavailability (3.4%, *n* = 3); staff unavailability (2.3%, *n* = 2); and other clinical reasons (1.1%, *n* = 1).
Retention Rate	>50% of enrolled patients remain in ICU for the full 10 days of study enrolment	45.6%	
Outcome Measure Assessments	100% of RFCSA ultrasound measurements were performed within 24 h of the scheduled time	100%	
	>75% of vascular, strength, and functional capacity measures were performed within 24 h of the scheduled time	80.8% of vascular measures 95.5% of strength measures 92.5% functional capacity measures	Rest vascular measures were not performed due to intolerance (11.5%, *n* = 3) and clinical reasons (7.7 %, *n* = 2). Rest strength and functional measures either performed after 24 h of schedule time (5%, *n* = 2) or not performed due to participant unavailability (2.5%, *n* = 1) or clinical reasons (2.5%, *n* = 1).
	>75% of surviving patients complete the quality-of-life questionnaires at 90-day follow-up	85.7%.	
Data Collection	<10% missing outcome data including ICU and hospital length of stay and survival	2.9%	
	<10% missing clinical data obtained from clinical medical notes and electronic patient records, such as the severity of illness scores and requirement for organ supportive therapies	<1%	

## Data Availability

The data generated from this study will be made available on request to the corresponding author.

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
