# Peer review of "Safety and Feasibility Assessment of Repetitive Vascular Occlusion Stimulus (RVOS) Application to Multi-Organ Failure Critically Ill Patients: A Pilot Randomised Controlled Trial"

_jcm, 2022, doi:10.3390/jcm11143938_

Round 1

Reviewer 1 Report

Major point; It would be desirable to publish the raw data of all measured cases in supplement data since the number of cases is quite small.

Minor points; What does this sentence 'Error! Reference source not found.' in Table 2 mean?

Reviewer 2 Report

Thank you for the opportunity to review the manuscript intitled “Safety and feasibility assessment of repetitive vascular occlusion stimulus (RVOS) application to multi-organ failure critically ill patients: a pilot randomised controlled trial”.

In this paper, the author conducted a pilot randomized controlled trial to assess the safety, tolerability and, feasibility of RVOS in ICU patients and to explore its impact on muscle, vascular function, and clinical outcomes.

The main findings were:

  • RVOS was an acceptable treatment for 75% participants and 82% clinical staff and was not associated with severe adverse event. However, 2 patients of 5 have not tolerated the treatment.
  • Intervention group was associated with less frequent AKI, a greater decrease in total SOFA score and increased IGF-1 and reduced syndecan-1, IL-4, and TNF-RII levels.
  • The authors concluded the application of RVOS to ICU patients appeared safe and acceptable, although not readily tolerated by all patients.

Findings are well described and the manuscript is appropriately organized.

My main concern is the small sample size in this study (only 12 patients of intended 32 recruited) which carries a considerable risk of bias in demonstrating the effectiveness or safety of the intervention.

One more time, thank you for the opportunity to review your manuscript.

Reviewer 3 Report

Chhetri I. et al investigated the effects of repetitive vascular occlusion stimulus (RVOS) on the muscle atrophy and vascular endothelial function in ICU patients. Indeed, it is a study with a well-structured methodology, despite the small sample size which may cause underpowered comparisons between the interventional and the control group.

There are some major points that need to be answered:

Abstract

  • Please define SAEs, AKI and all abbreviations
  • How about control group? Comparison to interventional group? Was there a statistical significance?
  • are required

Introduction

  • lines 65-67: there are studies (Hickmann CE, Castanares-Zapatero D, Deldicque L, Van den Bergh P, Caty G, Robert A, Roeseler J, Francaux M, Laterre PF. Impact of Very Early Physical Therapy During Septic Shock on Skeletal Muscle: A Randomized Controlled Trial. Crit Care Med. 2018;46:1436-1443) that have shown that early physical therapy is safe and preserves muscle fiber cross-sectional area in ICU patients. Please include it as a fact.

Methods

  • lines 138-142: How did you get an informed consent from mechanical ventilated patients in order to include them in the study? You refer that the declaration 139 of agreement was sought from the patient’s ‘Personal Consultee’ who was a representa-140 tive, partner, or close friend. But is that ethically right? Informed consent should be provided only by patient’s relatives. Moreover, obtaining informed consent from the patient afterwards is also not correct.
  • Line 144: Why did you choose right proximal lower limb? Did you have any data to support this? If not, please refer that it was an arbitrary choice.
  • Line 151: You refer that the maximum pressure used was 200 mmHg. I suppose that all patients had BP <150mmHg before the occlusion. Is that right?
  • Line 161: You assessed your parameters at day 1, 6 and 11 of study. Did you have any criterion for this choice? For example, why not day 1, 5 and 10?

Results

  • Table 1: Was there significant statistical difference of demographics between the 2 groups? Please include p value if significant.
  • You compared rectus femoris cross sectional area and echogenicity between right and left leg in 3 patients which is a quite small size and therefore, it is difficult to find differences. However, your p value was <0.05. Please provide the power of this analysis and the effect size. Same in Table 4 between interventional and control group.
  • Table 5: ICU Mobility score was higher in control group compared to interventional group at hospital discharge. Please explain.
  • Table 6: Superficial femoral artery outcome measures in control and intervention participants did not differ significantly. However, there is a tense for improvement in interventional group. Your sample size is small and this may be the reason for no differences. Please refer it in the text.
  • Supplementary table “Vascular and Inflammatory biomarkers levels”: Please provide p value for the comparisons between groups for each marker.

Round 2

Reviewer 1 Report

My concerns have been remedied.

Author Response

Thank you for reviewing our manuscript. 

Reviewer 2 Report

One more time, other studies have already reported similar results and i don’t think this study add sufficient novelty to thé Research field to be published in the journal.

thanks again 
